# Hybrid seed incompatibility in *Capsella* is connected to chromatin condensation defects in the endosperm

**Katarzyna Dziasek**[1☯], **Lauriane Simon**[1☯], **Clément Lafon-Placette**[1,2], **Benjamin Laenen**[3], **Cecilia Wärdig**[1], **Juan Santos-González**[1], **Tanja Slotte**[3], **Claudia Köhler**[1]*

**1** Department of Plant Biology, Uppsala Biocenter, Swedish University of Agricultural Sciences, Linnean Center of Plant Biology, Uppsala, Sweden, **2** Present address: Department of Botany, Faculty of Science, Charles University, Prague, Czech Republic, **3** Department of Ecology, Environment and Plant Sciences, Science for Life Laboratory, Stockholm University, Stockholm, Sweden

☯ These authors contributed equally to this work.
* Claudia.kohler@slu.se

**Data Availability Statement:** Data available from NCBI database. For the copy number analysis, we used data available under ERR636163, ERR636164, SRR8394203, SRR8394204 for *Cr*,

## Abstract

Hybridization of closely related plant species is frequently connected to endosperm arrest and seed failure, for reasons that remain to be identified. In this study, we investigated the molecular events accompanying seed failure in hybrids of the closely related species pair *Capsella rubella* and *C. grandiflora*. Mapping of QTL for the underlying cause of hybrid incompatibility in *Capsella* identified three QTL that were close to pericentromeric regions. We investigated whether there are specific changes in heterochromatin associated with interspecific hybridizations and found a strong reduction of chromatin condensation in the endosperm, connected with a strong loss of CHG and CHH methylation and random loss of a single chromosome. Consistent with reduced DNA methylation in the hybrid endosperm, we found a disproportionate deregulation of genes located close to pericentromeric regions, suggesting that reduced DNA methylation allows access of transcription factors to targets located in heterochromatic regions. Since the identified QTL were also associated with peri-centromeric regions, we propose that relaxation of heterochromatin in response to interspecies hybridization exposes and activates loci leading to hybrid seed failure.

## Author summary

Seed failure in response to interspecific hybridizations is a well-known reproductive barrier preventing interbreeding of closely related species and thus maintaining species boundaries. This reproductive barrier is established in the endosperm, a nourishing tissue supporting embryo growth. In this study, we discovered that the endosperm of interspecific hybrids between the recently diverged species *Capsella rubella* and *C. grandiflora* suffers from mitotic abnormalities and random chromosome loss. We found that the endosperm has reduced levels of DNA methylation and chromatin condensation, likely accounting for the chromosome loss. Importantly, we found that genes located in

and SRR5988314, SRR5988315, SRR5988316, SRR5988317 for *Cg*. Whole genome resequencing data from the *Cg* × *Cr* F1 and the *Cr* parent are available under PRJEB9020. Endosperm genomic DNA from *Cr*, *Cg*, and *Cr* × *Cg* and bisulfite sequencing data are available under PRJNA647289. Capsella endosperm RNA-seq data are available under GSE67359. Arabidopsis endosperm RNA-seq data are available under GSE84122.

**Funding:** This research was supported by grants from the Swedish Research Council VR (to CK, grant #2017-04119), a grant from the Knut and Alice Wallenberg Foundation (to CK, grant #2018-0206), and support from the Göran Gustafsson Foundation for Research in Natural Sciences and Medicine (to CK). The work of BL and TS was supported by grants from the Science for Life Laboratory and the Swedish Research Council (grant #621-2010-5508 and #621-2013-4320). Computational work was enabled by resources provided by the Swedish National Infrastructure for Computing (SNIC) at UPPMAX partially funded by the Swedish Research Council through grant agreement no. 2016-07213. The funders had no role in study design, data collection and analysis, decision to publish, or preparation of the manuscript.

**Competing interests:** The authors have declared that no competing interests exist.

pericentromeric regions were preferentially deregulated, suggesting that reduced DNA methylation exposes transcription factor binding sites in pericentromeric regions, leading to hyperactivation of genes and seed arrest. In support of the relevance of pericentromeric regions for hybrid seed arrest, we identified three QTL connected with the phenotype that were all located in pericentromeric regions. These results link epigenetic changes in hybrid endosperm with distinct genetic loci underpinning hybrid seed failure.

## Introduction

The endosperm is a nourishing tissue supporting embryo growth, similar to the placenta in mammals. In most flowering plants the endosperm is a triploid tissue, derived after fertilization of the diploid central cell by one of the haploid sperm cells [1]. The endosperm is sensitive to parental genome dosage and the ratio of two maternal to one paternal genome copies is required for normal development [2,3]. In most flowering plants the endosperm initially develops as a coenocyte, where nuclear divisions are not followed by cell wall formation [4]. The transition to cellularization is essential for viable seed formation [5]; however, the precise mechanism underlying this transition and the reason for its requirement remain to be identified. Hybridizations of plants that differ in ploidy disrupt the parental genome balance and affect endosperm cellularization, leading to seed arrest [6–9]. Similarly, hybridization between closely related species frequently leads to defects in endosperm development [6,8,10–13], suggesting similar underlying mechanisms. In support of this notion, interspecies hybridization barriers can be overcome by ploidy manipulations [7,14,15]. Despite the widespread occurrence of hybrid seed failure among flowering plants, the underlying molecular mechanisms are poorly understood.

In the *Capsella* genus, the selfing species *C. rubella* (referred to as *Cr*) separated less than 200,000 years ago from the obligate outcrosser *C. grandiflora* (referred to as *Cg*) [16–18]. Both species have the same ploidy level, but are separated by a strong endosperm-based hybridization barrier in both directions of hybridization [19,20]. Nevertheless, hybrid seed phenotypes differ depending on the direction of hybridization; when *Cg* maternal plants are pollinated with *Cr* pollen (referred to as *Cg* × *Cr*), the endosperm cellularizes precociously, giving rise to very small seeds. In the reciprocal cross (*Cr* × *Cg*), endosperm cellularization is delayed and seeds collapse [19]. Very similar reciprocal phenotypes result from interploidy crosses of *Arabidopsis*; seeds derived from crosses of tetraploid maternal plants with diploid pollen donors (4x × 2x) resemble *Cg* × *Cr* seeds, while crosses of diploid maternal plants with tetraploid pollen donors (2x × 4x) resemble *Cr* × *Cg* seeds [19,21]. Interploidy seed failure of 2x × 4x crosses is causally connected to increased expression of imprinted paternally expressed genes (PEGs) [22–25]. The PEG *PHERES1* (*PHE1*) encodes an AGAMOUS-LIKE (AGL) transcription factor that is also highly upregulated in triploid seeds [26,27]. PHE1 acts upstream of many PEGs, likely accounting for increased expression of PEGs and other direct PHE1 targets in triploid seeds, leading to seed arrest [25]. Many AGLs, including *PHE1* and *PHE1* orthologs, are also upregulated in *Arabidopsis* and *Capsella Cr* × *Cg* interspecies hybrids [19,28], correlating with the similar phenotypes of interploidy (2x × 4x) and interspecies hybrid seeds [14,19,21,28]. The endosperm derived from interploidy (2x × 4x) crosses has reduced levels of CHH methylation (H corresponds to A, T, or C), a hallmark of the RNA-directed DNA methylation (RdDM) pathway. The RdDM pathway establishes DNA methylation in all sequence contexts and is guided by 24-nt small RNAs (sRNAs) [29–31]. Activity of the RdDM pathway is low in the early endosperm and increases during endosperm development [32,33]. Maternal tissues

surrounding the female gametophyte form 24-nt sRNAs that accumulate in the endosperm [34,35] and may guide *de novo* methylation in the endosperm.

To elucidate the molecular cause of seed abortion in *Cr × Cg* interspecies hybrids, we performed a QTL analysis and identified three phenotype-associated QTL that were all localized in pericentromeric regions. Hybrid endosperm had strongly reduced CHG and CHH methylation, associated with chromatin decondensation, mitotic abnormalities and random chromosome loss. Deregulated genes in hybrids were preferentially localized in pericentromeric regions and enriched for orthologs of PHE1 targets, suggesting that increased expression of AGLs in hybrid endosperm ectopically activates target genes in hypomethylated pericentromeric regions.

## Results

### QTL associated with *Cr × Cg* incompatibility are localized in pericentromeric regions

To identify the genetic loci involved in hybrid seed incompatibility between *Cr* and *Cg*, we generated an F2 *Cg/Cr* population and crossed 480 F2 individuals as pollen donors to *Cr* maternal plants. We scored the seed abortion rate of the resulting hybrid seeds and genotyped the F2 individuals using a double digest restriction-site associated DNA (ddRAD) approach. This information was used for mapping quantitative trait loci (QTL) associated with hybrid seed abortion (Fig 1A). We detected three significant QTL located on chromosomes 2, 3 and 7

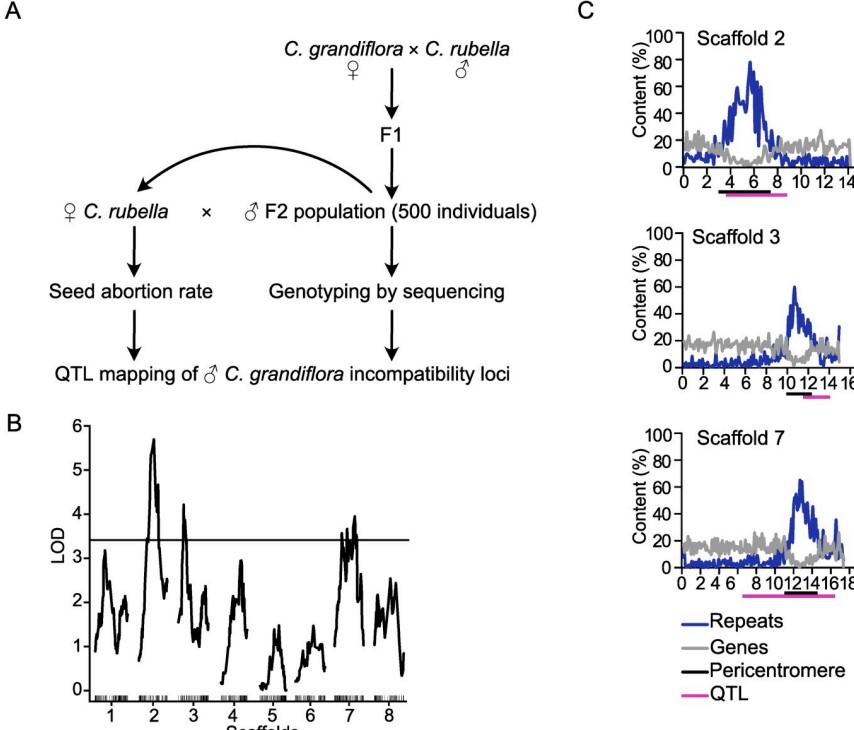

**Fig 1. QTL associated with *Cr × Cg* incompatibility are localized in pericentromeric regions.** A) Scheme of experimental design for QTL mapping. B) LOD scores for *C. grandiflora* QTL associated with abortion of *Cr × Cg* seeds. The plot shows the QTL LOD profile across the eight linkage groups corresponding to main pseudochromosomal scaffolds of the *C. rubella* genome assembly. The horizontal line represents the genome-wide significance threshold (= 0.05, estimated by 1000 permutations). C) Plots of gene (grey line) and repeat (blue line) content on three scaffolds containing identified QTL (from top to bottom: scaffolds 2, 3 and 7). The magenta bars represent QTL regions and the black bars indicate pericentromeric regions as previously defined [36].

(Fig 1B). All identified regions were very broad, ranging from positions 3.6 to 8.8 Mb on scaffold 2, 11.5 to 14.1 Mb on scaffold 3, and 6.4 to 16.4 Mb on scaffold 7. The strong QTL on scaffold 2 overlaps with a previously identified QTL based on a *Cg/Cr* RIL population, while we did not identify the weak QTL previously found on scaffold 8 [19]. Strikingly, all three QTL span centromeric and pericentromeric regions [36], suggesting a role of centromeric/pericentromeric regions in hybrid incompatibility in *Capsella*.

## Hybrid endosperm shows mitotic abnormalities and random chromosome loss

Previous work revealed that interspecific hybrids frequently encounter chromosome loss, generally resulting in the complete elimination of one parental genome [37–39]. This uniparental genome elimination is caused by the loss of the centromere-specific histone H3 variant CENH3 from one of the parental genomes in the hybrid [40]. We hypothesized that the presence of the three QTL in centromeric/pericentromeric regions may be connected to chromosome loss in *Capsella* hybrid endosperm. To test this hypothesis, we counted the number of chromocenters in nuclei of hybrid and parental endosperms. We found indeed that the hybrid endosperm had at least one chromocenter less compared to the parental species (Fig 2A and 2B). The shape of the chromocenters was strikingly different between parents and hybrid; while *Cr* and *Cg* interphase endosperm nuclei had clearly formed chromocenters, in the hybrid endosperm the shape of the chromocenters was diffuse, indicating that the chromatin was less condensed (Fig 2B). Close inspection of metaphase plates in hybrid endosperm nuclei revealed the presence of abnormally shaped spindles and misaligned chromosomes, which were not observed in the endosperm of the *Cr* parental species (Fig 2C).

The chromocenters in embryos were highly dispersed, making it not possible to count the number of chromocenters directly in the embryos of the hybrid seeds. In order to test whether the observed defects were endosperm specific, we rescued hybrid *Cr* × *Cg* embryos and performed chromosome spreading from root nuclei of the obtained hybrid plants. Neither the number of chromocenters (Fig 2D) nor the chromatin condensation differed between parental and hybrid root tissue (Fig 2E), revealing that chromosome loss was restricted to the endosperm.

We next addressed the question whether there was a specific chromosome that was lost in the hybrid endosperm, or whether the loss occurred randomly. We manually dissected endosperm from hybrid and the non-hybrid seeds, isolated DNA and performed high-throughput sequencing. We then determined the coverage over all 8 scaffolds of the *Cr* genome (Fig 2F), but did not identify a significant coverage reduction in any scaffold, as expected if a specific chromosome would be lost. We confirmed this observation by plotting the proportion of *Cg* to total SNPs (single nucleotide polymorphisms) in all scaffolds. Since *Cg* was the paternal parent in the hybrid endosperm, this ratio is expected to be 33.3%. The obtained ratio was close to 40% in all scaffolds and did not significantly differ between the scaffolds, consistent with the similar coverage over all scaffolds (Fig 2F). Together, this data strongly suggest that the hybrid *Cr* × *Cg* endosperm suffers from random chromosome loss and that chromosome loss is specific for the hybrid endosperm and does not occur in the hybrid embryo.

## Characterization of centromeric repeats of *Cr* and *Cg*

The outcome of the QTL mapping together with the fact that the hybrid endosperm encountered random chromosome loss, prompted us to investigate whether the centromeric regions of *Cr* and *Cg* differ. Fluorescence in situ hybridization (FISH) signals obtained with probes of the *Cr* centromeric consensus repeats colocalized with the chromocenters in *Cr* and *Cg* (Fig

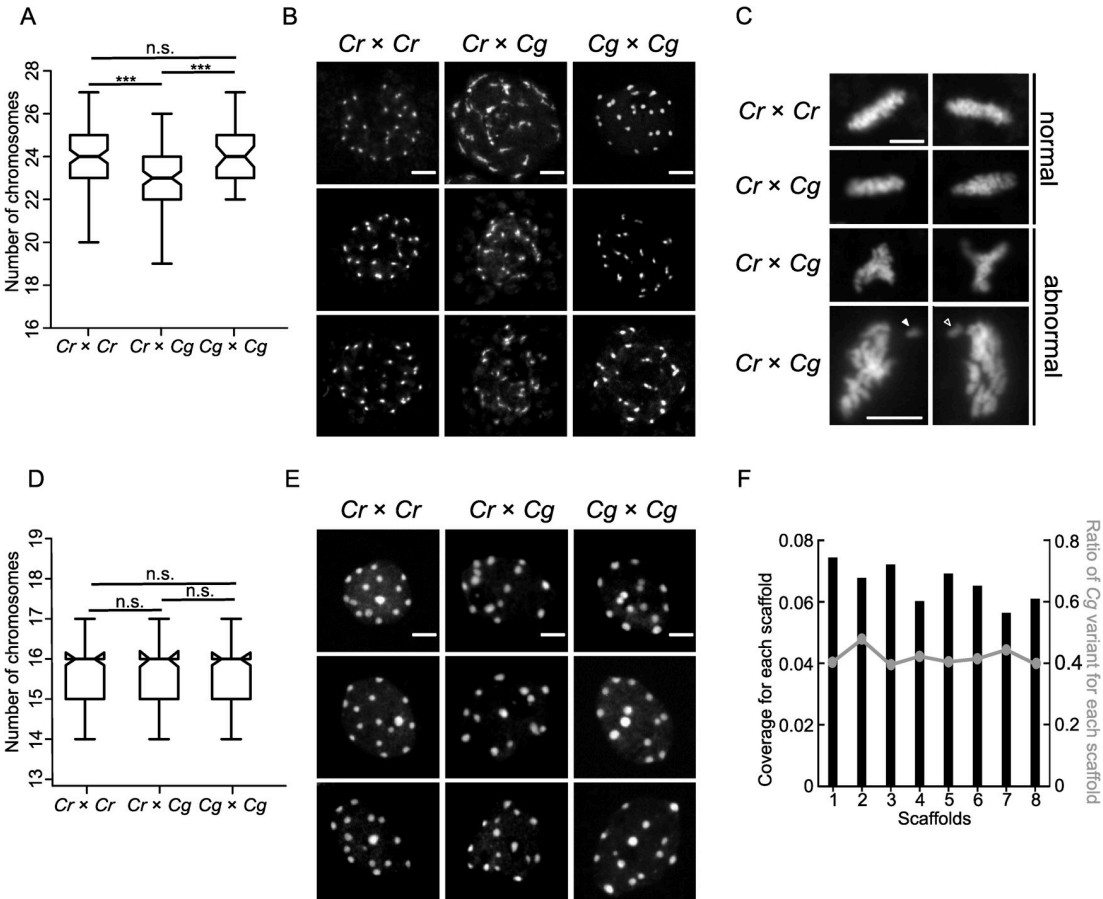

**Fig 2. Hybrid endosperm shows random chromosome loss and mitotic abnormalities.** A) Boxplots showing number of chromosomes in nuclei of $Cr \times Cr$ (n = 101), $Cr \times Cg$ (n = 101) and $Cg \times Cg$ (n = 53) 4 DAP seeds. Boxes show medians and the interquartile range, and error bars show the full range excluding outliers. Asterisks indicate significant differences calculated by Wilcoxon test (*** p-value < 0.001). B) DAPI stained chromocenters from endosperm nuclei of $Cr \times Cr$, $Cr \times Cg$ and $Cg \times Cg$. Scale bar, 5 μm. C) DAPI stained pictures of metaphase plates of $Cr \times Cr$ and $Cr \times Cg$ endosperm nuclei from 4 DAP seeds. Arrows indicate lagging chromosomes. Scale bar, 5 μm. D) Box plots showing number of chromosomes in root nuclei of $Cr \times Cr$, $Cr \times Cg$ and $Cg \times Cg$ seedlings. For each genotype 100 nuclei were analyzed. Boxes show medians and the interquartile range, and error bars show the full range excluding outliers. Differences between genotypes are not significant (Wilcoxon test). E) DAPI stained chromocenters from root nuclei of $Cr \times Cr$, $Cr \times Cg$ and $Cg \times Cg$ seedlings. Scale bar, 3 μm. F) Histogram representing the coverage of each scaffold in hybrid nuclei. The line represents the ratio of $Cg$ to total reads in each scaffold in hybrid seeds.

3A), indicating that centromeric repeats of $Cr$ and $Cg$ were similar and positioned in the centromeric region.

We tested whether there are species-specific SNPs for $Cr$ or $Cg$ in the centromeric region. However, in the published centromeric $Cr$ sequence [41] we did not identify specific mutations present only in one of the two species, the SNP profiles for $Cr$ and $Cg$ centromeric repeats were almost identical (S1 Fig).

To address the question whether the number of centromeric repeats differ between $Cr$ and $Cg$, we blasted the consensus sequence of the $Cr$ centromeric repeats [41] to $Cg$ scaffolds and found 88 scaffolds with multiple hits organized in tandem repeats, revealing that the $Cg$ genome contains centromeric repeats similar to $Cr$. Next, we used the $Cr$ centromeric consensus sequence as reference and mapped the publicly available genomic data of four different $Cg$ and $Cr$ populations to the centromeric repeat sequence. On average, 8.9% of the $Cg$ reads could be mapped to centromeric repeats, while only an average of 6.7% of the $Cr$ reads,

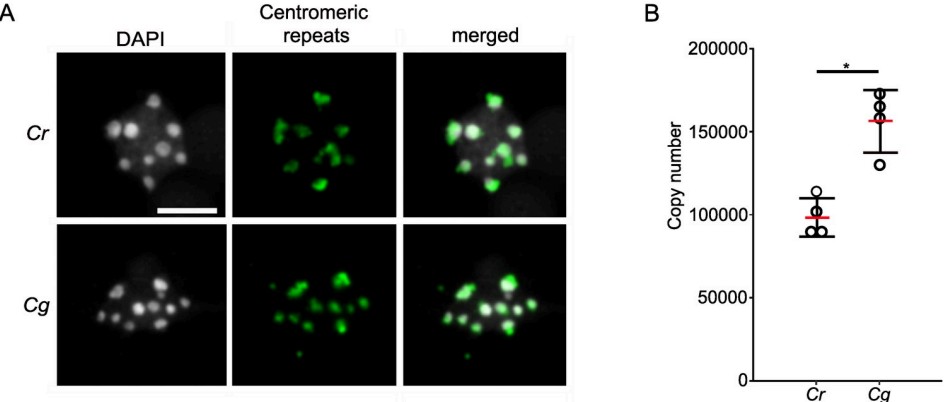

**Fig 3. Characterization of centromeric repeats of *Cr* and *Cg*.** A) FISH with centromeric repeat probes on *Cr* and *Cg* leaf nuclei. Scale bar, 5 μm. B) Column scatter plot showing copy number of centromeric repeats in *Cr* and *Cg* genomes. Red bar represents the mean value. Asterisks indicate significant differences calculated by Wilcoxon test (* p-value < 0.05).

revealing that centromeric repeats are more abundant in the *Cg* compared to the *Cr* genome. To further test this, we determined the centromeric repeat copy number in *Cr* and *Cg* by comparing the coverage of the centromeric repeats with the coverage of single copy genes. We found indeed about 1.5 times more copies in *Cg* (~150 000) than in *Cr* (~100 000) (Fig 3B).

Together, these results show that centromeric repeats from *Cr* and *Cg* are highly similar in sequence, but that the *Cg* genome contains substantially more copies of centromeric repeats than the *Cr* genome.

## Hybrid endosperm shows chromatin decondensation associated with loss of DNA methylation on transposable elements

The centromere-specific histone variant CENH3 is crucial for proper chromocenter formation [42]. Unlike typical histones, CENH3 is a fast evolving protein and CENH3 differences between species were reported to cause chromosome elimination [40,43,44]. Alignment of the protein sequences of CENH3 from *Cr* (Carubv10010436m) and *Cg* (Cagra.1968s0080.1) revealed that they differed in two amino acids, while the canonical H3 histones in these two species were identical (S2 Fig). In order to test whether there were global changes in CENH3 loading in the hybrid endosperm nuclei in comparison to non-hybrid nuclei, we generated an antibody against *Capsella* CENH3 and performed immunolocalization using this antibody. However, we did not observe obvious differences in CENH3 localization between parental *Cr* and hybrid endosperm (Fig 4A), indicating that CENH3 is properly loaded in the hybrid endosperm. We therefore considered it unlikely that failure of CENH3 loading accounts for the observed chromosome loss.

Chromocenters of hybrid endosperm were less condensed compared to the parental endosperm (Fig 2B). Since loss of DNA methylation is frequently accompanied with reduced chromatin condensation and heterochromatin loss [45,46], we tested whether reduced DNA methylation in hybrid endosperm may account for reduced chromocenter condensation, leading to chromosome loss. We performed bisulfite sequencing of DNA isolated from parental and hybrid endosperm at 4 DAP. We observed a strong decrease of CHG and CHH methylation on transposable elements (TEs) in hybrid endosperm in comparison to both parental species (Fig 4B and 4C); while genes had an intermediate level of DNA methylation in all sequence contexts compared to the parental species (S3 Fig).

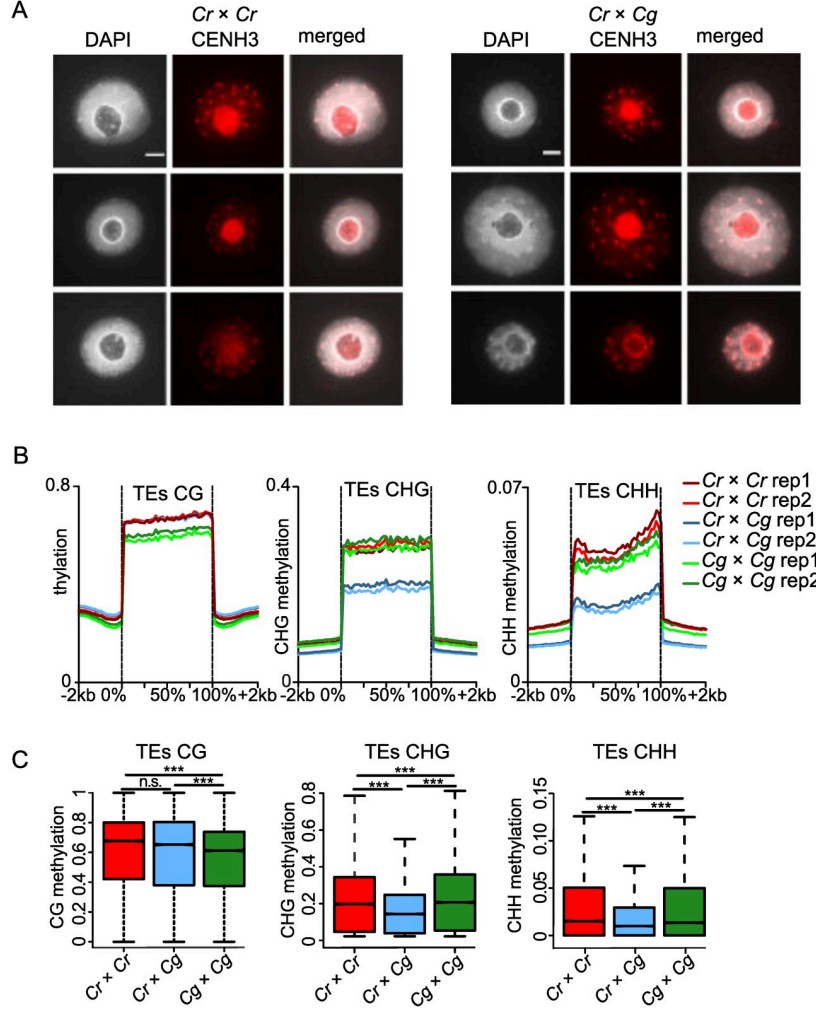

**Fig 4. Endosperm chromatin decondensation associates with loss of DNA methylation but not CENH3 occupancy.** A) Immunostaining of CENH3 in the endosperm of *Cr × Cr* and *Cr × Cg* 4 DAP seeds and chromatin staining with DAPI. Scale bar, 3μm. B) Methylation level of CG, CHG and CHH of transposable elements (TEs) in *Cr*, *Cr × Cg* and *Cg* endosperm of 4 DAP seeds. C) Boxplots showing the methylation level of transposable elements (TEs) in *Cr × Cr* and *Cg × Cg* and *Cr × Cg* endosperm of 4 DAP seeds. Boxes show medians and the interquartile range, and error bars show the full range excluding outliers. Asterisks indicate statistically significant differences calculated by Wilcoxon test (*** p-value < 0.001, * p-value < 0.05).

## Deregulated genes in hybrid endosperm are preferentially localized in pericentromeric regions

Around half of the TEs losing DNA methylation were localized in pericentromeric regions (S1 Table), which was significantly more than expected by chance (p<0.0001, Chi-square test). We therefore asked whether deregulated genes are also preferentially located in pericentromeric regions. We analyzed previously published transcriptome data of parental and hybrid seeds [19] to test the spatial distribution of genes that were overexpressed in the hybrid relative to the *Cr* parent. For this purpose, we divided each scaffold into deciles containing equal numbers of genes and plotted the number of upregulated genes in each group in all scaffolds (Fig 5A). Strikingly, upregulated genes were preferentially localized in pericentromeric regions (determined in [36] in all scaffolds (p< 0.0001, binomial test), suggesting that loss of DNA methylation and chromatin decondensation cause preferential activation of genes in

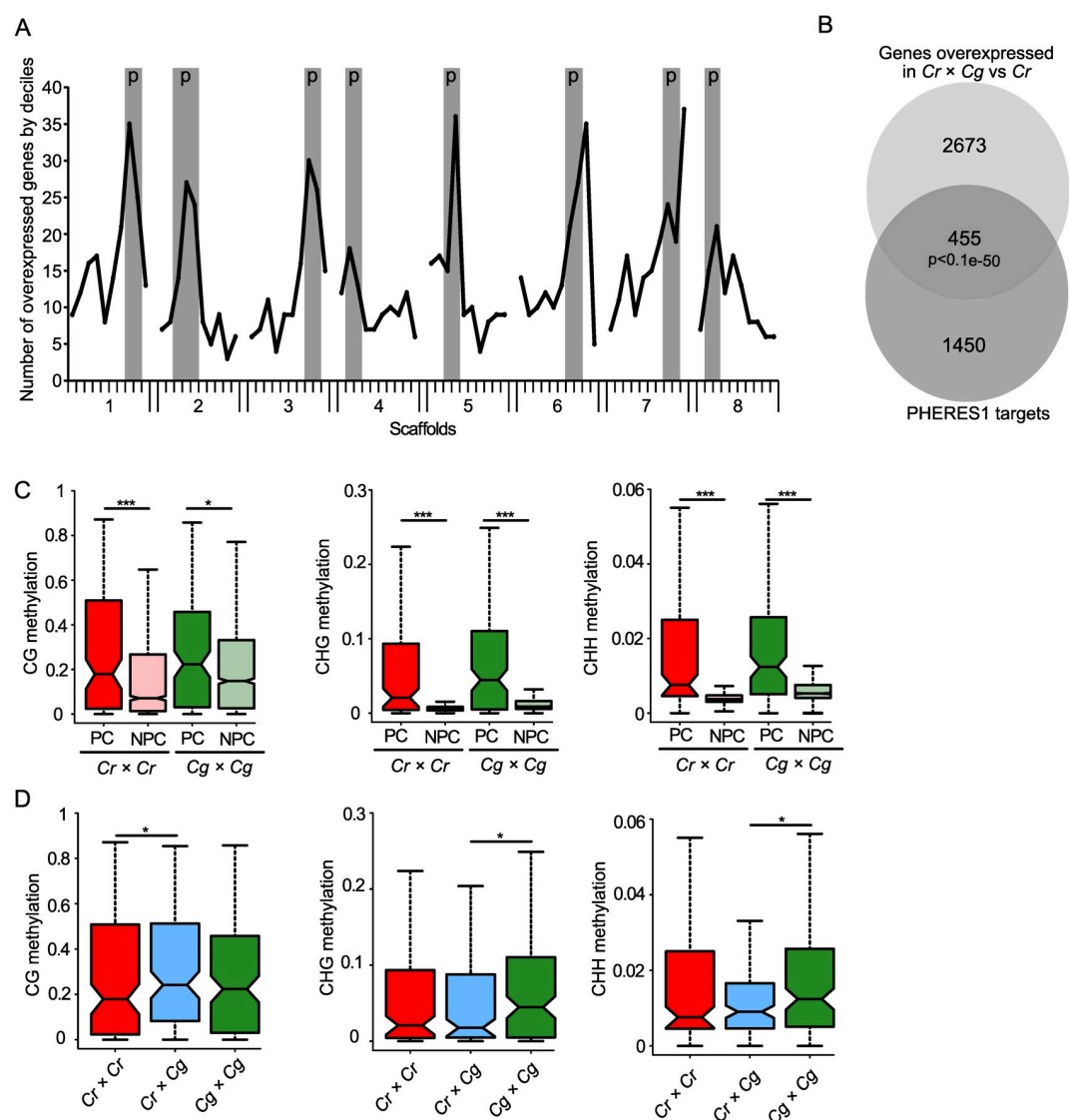

**Fig 5. Deregulated genes in hybrid endosperm are preferentially localized in pericentromeric regions.** A) Number of genes overexpressed in *Cr × Cg* in comparison to *Cr* per decile of genes on each scaffold. Pericentromeric regions (p) are highlighted in grey for each scaffold. B) Venn diagram showing the overlap between significantly overexpressed genes in hybrid seeds compared to *Cr × Cr* and PHE1 targets. C) Boxplots showing average methylation levels of the promoter regions (upstream 500 bp of the transcriptional start site) of PHE1 target orthologs in pericentromeric (PC) and non-pericentromeric (NPC) regions in *Cr × Cr* and *Cg × Cg* endosperm of 4 DAP seeds. D) Boxplots showing methylation levels of the promoter regions of PHE1 target orthologs in pericentromeric regions in *Cr × Cr*, *Cr × Cg*, and *Cg × Cg* endosperm of 4 DAP seeds. Boxes show medians and the interquartile range, and error bars show the full range excluding outliers. Asterisks indicate significant differences calculated by Wilcoxon test (*** p-value < 0.001, ** p-value < 0.01, * p-value < 0.05).

pericentromeric regions. We found a similar pattern, while less pronounced, when analyzing genes being deregulated in hybrid seeds in comparison to both parents. Also here, there was a preferential localization of upregulated genes in pericentromeric regions (S4A Fig) (p< 0.0001, binomial test). The pattern for downregulated genes looked strikingly different, where with the exception of scaffold 1 no enrichment in pericentromeric regions was detected (S4B Fig).

We previously found that AGAMOUS-LIKE (AGL) MADS-box transcription factors are strongly upregulated in *Capsella* hybrid endosperm [19]. Similarly, AGL transcription factors

are also highly upregulated in triploid *Arabidopsis* seeds [5,26,28,47] and the AGL transcription factor PHERES1 (PHE1) was shown to causally account for triploid seed arrest [25]. The *Capsella* ortholog of *PHE1* (Carubv10020903m.g, referred to as *CrPHE1*) is also highly upregulated *Capsella* hybrid seeds [19]. We therefore tested whether deregulated genes in hybrid *Capsella* seeds were enriched for orthologs of PHE1 target genes. We found that out of the 3128 upregulated genes in the hybrid compared to *Cr*, 455 were orthologs of PHE1 targets (Fig 5B), which is a significant overlap (p<1.e-50, hypergeometric test). Interestingly, there were also significantly more pericentromeric PHE1 targets (26 out of 146 (= 17.8%)) overexpressed in hybrid seeds compared to non-pericentromeric PHE1 targets (213 out of 1759 (= 12.1%)) (p<0.05, hypergeometric test). The overlap of PHE1 targets with deregulated genes was also significant when testing deregulated genes in the hybrid in comparison to both parents (S4C Fig; p<0.05, hypergeometric test).

Previous work suggested that PHE1 binding is prevented by DNA methylation [25], suggesting that increased expression of PHE1 targets in pericentromeric regions of hybrid endosperm is a consequence of reduced DNA methylation in those regions. We compared the DNA methylation levels in the promoter region of PHE1 target orthologs in pericentromeric regions and non-pericentromeric region and found indeed that targets in pericentromeric regions had higher levels of DNA methylation in all sequence contexts compared to non-pericentromeric targets (Fig 5C). While CG methylation of pericentromeric targets in the hybrid endosperm was slightly increased compared to parental methylation, CHG and CHH methylation was significantly lower in the hybrid compared to parental *Cg* endosperm (Fig 5D). Importantly, the PHE1 target motifs predominantly contain cytosines in CHH context (Batista et al., 2019) [25]; supporting the idea that loss of CHH methylation in hybrid endosperm may expose binding sites for CrPHE1 and cause increased expression of CrPHE1 targets. Among the deregulated orthologs of PHE1 targets were *AGL40* on scaffold 7 and *AGL95* on scaffold 3, located within the identified QTL. Based on yeast-two-hybrid interaction data, AGL40 encodes for a direct interaction partner of PHE1; while AGL95 is a paralog of PHE1 [48]. Both genes were also highly upregulated in triploid seeds (S5 Fig), as expected for direct PHE1 target genes. There was a pronounced loss of CHH and CHG methylation in the promoter and coding region of *AGL40* in the hybrid endosperm compared to both parents (S5 Fig). *AGL95* also had reduced DNA methylation in the hybrid endosperm compared to the *Cg* parent (S5 Fig). Moreover, other PHE1 targets that were deregulated in *Capsella* hybrids were similarly deregulated in triploid seeds (S6 Fig).

Together, this data support the idea that loss of DNA methylation in hybrid endosperm exposes binding sites for CrPHE1 and potentially other type I AGLs and the resulting increased expression of those targets causally connects to failure in endosperm cellularization and seed arrest.

## Discussion

Understanding the molecular cause for hybrid incompatibility is a major goal of evolutionary biology. It is also of high relevance for plant breeding, since it may facilitate the generation of new hybrid varieties. In this study, we provide insights into hybrid incompatibility of two closely related *Capsella* species. We found that *Cr* × *Cg* hybrid seeds undergo endosperm-specific chromatin decondensation, leading to random chromosome loss. Chromatin decondensation is likely a consequence of reduced DNA methylation in the endosperm. Hypomethylation in pericentromeric regions exposes binding sites for the AGL transcription factor CrPHE1, leading to hyperactivation of potential CrPHE1 targets.

## Pericentromeric regions play a role in hybrid incompatibility

To uncover the genetic elements involved in *Cr* × *Cg* incompatibility, we performed QTL mapping using an F2 *Cg*/*Cr* population. In a previous study we followed a similar approach using *Cg*/*Cr* recombinant inbred lines (RILs) [19]. However, the majority of RILs did not trigger seed abortion when crossed with *Cr*, indicating that the alleles responsible for incompatibility are purged from the RIL population. To overcome this problem, in this study we used a *Cg*/*Cr* F2 population and identified three *Cg* loci that contribute to *Cr* × *Cg* incompatibility, consistent with our previous genetic prediction [19]. One of the QTLs on scaffold 2 was also detected in our previous study [19]. Interestingly, all three QTL were localized in pericentromeric regions, suggesting a particular role of pericentromeric regions in hybrid incompatibility. Pericentromeric heterochromatic regions were previously shown to be relevant for hybrid incompatibility in Drosophila. Hybrid incompatibility genes *Lhr* (*Lethal hybrid rescue*) and *Hmr* (*Hybrid male rescue*) encode for heterochromatin proteins and localize to centromeric heterochromatin [49–51]. Furthermore, the incompatibility locus *Zhr* (*Zygotic hybrid rescue*) is a species-specific heterochromatic repeat present on the X-chromosome of *D. melanogaster* but absent in *D. simulans* [52]. The paternal *D. melanogaster* X chromosome, containing the *Zhr* repeats, fails to segregate in a *D. simulans* maternal background. Our finding that the three *Cr* × *Cg* incompatibility QTL mapped to pericentromeric regions and that hybrid endosperm underwent chromosome loss, prompted us to investigate the role of pericentromeric regions in hybrid incompatibility. Previous work revealed that hybridization of *Arabidopsis* lines expressing species-specific variants of CENH3 to wild-type individuals causes severe chromosome segregation errors [44]. These abnormalities occurred in embryo and endosperm, contrasting to the endosperm-specific chromosome loss that we observed in *Cr* × *Cg* hybrids. Adding the fact that we did not observe any differences in CENH3 loading on mitotic chromosomes in hybrid endosperm, we consider it unlikely that the two amino acid differences between *Cr* and *Cg* CENH3 account for the observed chromosome loss. Our data rather point that differences in pericentromeric repeat number between *Cr* and *Cg* connects to hybrid seed failure. We found that length and sequence composition of *Cg* centromeric repeats was similar as previously described for *Cr* [41]. However, centromeres of *Cg* contained about 50% more repeats than *Cr*, revealing species-specific differences in centromere size between *Cr* and *Cg*. The precise size of the *Cg* genome is unknown, but flow cytometry analysis predicts that it is about 10% larger than *Cr* [53]. Based on our work, this difference is likely to be contributed by the increased number of centromeric repeats of *Cg*.

## Possible consequences of differential centromeric repeat numbers in *Cr* and *Cg*

Maternal tissues are the main source of 24-nt sRNAs accumulating in the endosperm [34,35] and likely guide *de novo* methylation in the endosperm. Since sperm DNA is highly depleted of CHH methylation [54,55], increased paternal genome dosage or, possibly, increased numbers of centromeric repeats, may lead to remethylation failure, if maternal 24-nt siRNAs are rate-limiting. Consistent with this scenario, CHH methylation is strongly depleted in the endosperm of 2x × 4x hybrid endosperm [56,57], similar to what we observed in this study. Centromeric repeats in *Arabidopsis* are also present in pericentromeric regions where they are heavily methylated [58]; therefore, increased numbers of centromeric repeats may impact on methylation levels outside of the centromere. Thus, non-matching dosage of maternal 24-nt sRNAs and paternal genome copies or repeats may lead to hypomethylation and consequently decondensation and random chromosome loss in the endosperm. The consistent phenotypes observed in *Cr* × *Cg* hybrid seeds make it however unlikely that random chromosome loss is

causally connected to endosperm failure. Both, $Cr \times Cg$ and 2x $\times$ 4x *Arabidopsis* hybrid seeds show similar phenotypic defects, most prominently cellularization defects in the endosperm [19,21]. Increased expression of the AGL transcription factor *PHE1* is causally responsible for 2x $\times$ 4x *Arabidopsis* hybrid seed defects [25] and *CrPHE1* and related AGLs are similarly highly upregulated in $Cr \times Cg$ hybrid seeds [19]. Consistent with the idea that increased expression of *CrPHE1* and related AGLs are connected to $Cr \times Cg$ hybrid seed failure, we found a high over-representation of PHE1 target orthologs among deregulated genes in *Capsella* hybrids. Importantly, PHE1 binding was shown to be negatively impacted by DNA methylation [25]. Since pericentromeric regions lose DNA methylation in hybrids, this may explain preferential activation of CrPHE1 targets in hypomethylated pericentromeric regions.

In summary, in this study we uncovered the molecular defects occurring in $Cr \times Cg$ hybrid endosperm and its likely genetic cause. We report that hybrid endosperm has reduced levels of CHG and CHH methylation, likely causing reduced chromatin condensation and random chromosome loss. We speculate that the cause for the hypomethylation is an imbalance of maternal *Cr* 24-nt sRNAs and *Cg* centromeric repeats. Increased expression of *CrPHE1* and related AGLs hyperactivate *CrPHE1* targets preferentially in hypomethylated pericentromeric regions, causing a phenotypic mimic to interploidy hybrid seeds in *Arabidopsis*.

## Materials and methods

### Plant material and growth conditions

In this study, we used the *Cr* accessions *Cr*48.21 and *Cr*1g and the *Cg* accessions *Cg*89.3, *Cg*81, *Cg*89.16 and *Cg*94. Seeds were surface sterilized and sown on agar plates containing ½ Murashige and Skoog (MS) medium and 1% sucrose. After stratification for 2 days in the dark at 4˚C, seedlings were grown in a growth room under long-day photoperiod (16 h light and 8 h darkness) at 22˚C light and 20˚C darkness temperature and a light intensity of 110 µE. Seedlings were transferred to pots and plants were grown in a growth chamber at 60% humidity and daily cycles of 16 h light at 21˚C and 8 h darkness at 18˚C and a light intensity of 150 µE.

### DNA isolation and preparation of ddRAD-Seq libraries

Mature leaves of $Cg \times Cr$ $F_2$ individuals (480 in total) were collected and flash frozen in liquid nitrogen. Genomic DNA was isolated from leaves using a CTAB extraction protocol [59]. We used a double-digest RAD-sequencing (ddRAD-seq) protocol [60] modified according to [61]. Briefly, about 500 ng of DNA per sample was successively digested with the restriction enzymes EcoRI and Taq$^{\alpha}$I. The resulting fragments were ligated with restriction site-specific barcoded adapters and size-selected (to ~550 bp) using AMPure beads (Beckman). The adapters were labeled with biotin, which allowed to perform an additional selection of adapter-ligated fragments using Dynabeads M-270 Streptavidin (Invitrogen). In total, 5 dual indexing 96-plex ddRAD-seq libraries were prepared of the $F_2$ samples. The $F_1$ and *C. rubella* parental samples were previously sequenced [62]. ddRAD-seq libraries were processed with 125bb paired-end sequencing on a total of five lanes (one library per lane) of Illumina HiSeq2500 system at the SNP&SEQ Technology Platform of SciLifeLab, Uppsala, Sweden.

### ddRAD-Seq read processing, variant calling and filtering

We obtained on average over 2.2 million reads per $F_2$ individua. Short-read data were demultiplexed and trimmed from barcode sequences using ipyrad [63] and we detected and trimmed sequencing adapters using trimmomatic v0.36 [64]. Trimmed reads were mapped to the v1.0 *Cr* reference genome assembly [18] using BWA-MEM [65]. We called variants and genotypes

using GATK 3.8–0 [66] HaplotypeCaller after Base Quality Score Recalibration (BQSR) using a set of known SNPs [67]. We filtered the resulting vcf file using VCFtools [68] to retain only biallelic SNPs that were genotyped in at least 95% of the samples with a read depth between 8 and 200 and a mapping quality of at least 50. SNPs in repetitive regions (identified using RepeatMasker as in [62] were further removed using bedtools v. 2.26.0 [69]. After filtering, we retained a total of 13,326 SNPs.

## Linkage map construction

We constructed a linkage map based on our SNP data in R/QTL [70]. For efficient linkage map construction, we first thinned our SNP set to retain 948 equally spaced SNPs using mapthin [71]. We inferred the parental origin of all SNP alleles based on whole-genome rese-quencing data from the *Cr* parent and the F1 [62]. We discarded individuals that were geno-typed for less than 95% of markers and filtered markers for segregation distortion as recommended in the R/QTL manual. The final map was constructed based on 623 markers genotyped in 383 F2s. Briefly, we partitioned SNPs into linkage groups using a maximum recombination fraction of 0.35 and a minimum LOD score of 8, and ordered markers on each linkage group. The resulting linkage map had a total length of 529 cM and eight linkage groups with more than one SNP marker, in good agreement with the expected haploid chromosome number ($n = 8$) of *Cr* and *Cg*.

## QTL mapping

The phenotyping was performed by crossing each F2 individual as pollen donor to the *Cr*48.21 accession as maternal plant, and for each *Cr* × F2 cross, 5 siliques were harvested, amounting to about 60 seeds per cross. The rate of aborted seeds (shriveled and dark) was measured for each cross and was used as continuous trait for the QTL analysis. The QTL analysis was done using interval mapping with the expectation maximization algorithm in R/QTL [70]. The background control parameter was set to a standard model with backward regression to select any possible QTL with standard five control markers. The QTL analysis gave three significant peaks that were located in three different chromosomes. The genome-wide significance thresh-old (alpha = 0.05) was obtained based on 1000 permutations. In the three significant QTLs, the presence of *Cg* alleles correlated with seed abortion.

## Embryo rescue

Embryo rescue was performed as described in [19].

## Chromosome spreading in root tips

Seedlings were grown on ½ MS agar plates. After 10 days, the root tips (about 1 cm) were cut with a razor blade. Roots were treated with colchicine (100 μM), 8-hydroxyquinoline (2.5 mM) and oryzalin (100 μM) for 2h at room temperature (RT) to block cell division and then fixed in ethanol:acetic acid (3:1) for 3h at RT. Fixed tissue was rinsed with citrate buffer for 10 min, transferred to 500 μl enzyme mix (0.3% w/v cytohelicase,0.3% w/v pectolyase, 0.3% w/v cellulase) and incubated at 37˚C for 1h. Then, the enzyme mix was removed and the tissue was washed with citrate buffer for 30 min. Root tips were incubated in a drop of 45% acetic acid for a few minutes and then transferred on a clean microscope slide. Tissue fragments were teased apart with a fine needle and gently squashed with the cover slip. Slides were frozen in liquid nitrogen; cover slips were removed and the squashed tissue was air dried. Slides were

mounted with Vectashield mounting medium with 4′,6-Diamidine-2′-phenylindole dihydrochloride (DAPI) (BioNordika AB).

## Immunostaining

10 siliques at 4 days after pollination (DAP) for each genotype were harvested and fixed in cold 4% formaldehyde in Tris buffer (10 mM Tris-HCl pH 7.5, 10 mM NaEDTA, 100 mM NaCl) for 20 min and washed for 2×10 min with cold Tris buffer. Seeds were isolated from the siliques and chopped with a razor blade in 100 µl LB01 buffer (15 mM Tris-HCl pH 7.5, 2 mM NaEDTA, 0.5 mM spermine, 80 mM KCl, 20 mM NaCl and 0.1% Triton X-100). The cell slurry was filtered through a 30 µm falcon cell strainer. 5 µl of nuclei suspension was mixed with 10 µl of sorting buffer (100 mM Tris-HCl pH 7.5, 50 mM KCl, 2 mM MgCl2, 0.05% TWEEN-20 and 5% sucrose), spread on a polylysine slide and air dried for 2 h. Slides were postfixed in 2% formaldehyde in PBS for 5 min and washed with water. Slides were covered with 1X PBS +0.5% Triton X-100 and kept in a moist chamber for 45 min at RT, then washed 3 times with 1X PBS for 5 min. Slides were denatured by adding 30 µl of deionized water and heated on a preheated plate at 80˚C for 8 min, then cooled down by dipping in 1XPBS. Slides were incubated with primary antibody (diluted 1:100 in 5% BSA, 0.05% TWEEN-20 in 1X PBS) for 1h at RT and then overnight at 4˚C, then washed the 3 times in 1XPBS 5min at RT and incubated with the secondary antibody (1:200, Abcam ab175471) for 2-3h at RT. Slides were washed 3 times with PBS for 5 min at RT and mounted with Vectashield mounting medium with DAPI (BioNordika AB). The experiment has been performed in three independent biological replicates.

## Endosperm nuclei spreading

4 DAP seeds were harvested and incubated overnight in a mix of 2.5 mM 8-hydroxyquinoline, 100 µM oryzalin, 100 µM colchicine, then fixed for at least 5h in fixative ethanol:acetic acid (3:1) at 4˚C. Seeds were washed with 10 mM citrate buffer and incubated for 5h in an enzyme mix containing 0.3% cytohelicase, 0.3% pectolyase, and 0.3% cellulase in 10mM citrate buffer. After digestion, 5~10 seeds were put on a slide, squashed with a needle, spread with acetic acid (60%) and fixed with 3:1 ethanol:acetic acid on the slide. Slides were mounted with Vectashield mounting medium with DAPI (BioNordika AB). The experiment has been performed in three independent biological replicates.

## Fluorescence in situ hybridization

For fluorescence in situ hybridization (FISH), three week old leaves were fixed in ethanol-acetic acid (3:1) and FISH was performed as described [72]. Centromeric repeat probes were amplified by PCR using Biotin-11-dUTP (Thermofisher) with primer TCTAGCACTTGTA ATCAATCAAATTC and AGAAGTGAGAAGAAAGACTTG. To detect the probe, an anti-Biotin antibody (FITC) (ab53469, Invitrogen) was used at a concentration of 1/1000.

## Copy number estimation

To estimate the copy number of centromeric repeats through next-generation-sequencing (NGS), we divided the average coverage over the repeats by the average coverage over three reference genes (Carubv10009577m, Carubv10011569m, Carubv10006001m for *Cr* and their orthologues Cagra.3392s0015, Cagra.0804s0001, Cagra.3807s0042 for *Cg*). For each individual dataset analyzed in this study, we mapped the reads to a single reference consisting of the *Cr*

centromeric repeat consensus sequence [41] using BWA-MEM (v0.7.8) [65]. We retrieved per-base read depth with the function Depthofcoverage from GATK (v3.5) [66].

### Identification of polymorphisms

To determine polymorphisms in centromeric repeats, we mapped the reads with BWA-MEM [65] to the consensus centromeric repeat of *Cr* and used previously published scripts (https://gist.github.com/laurianesimon/a9fc44aa83305c576e914710cae75f87#file-listsnpmodules_v2; https://gist.github.com/laurianesimon/a9fc44aa83305c576e914710cae75f87#file-countpolymorphisms_v4 [73]) and extracted and quantified polymorphisms for each position of the centromeric repeat from all mapped reads. We used whole genome sequencing data of leaves SRR5988314 and ERR636164 for *Cg* and *Cr* polymorphisms, respectively.

### Whole genome and bisulfite sequencing

Endosperm from 4DAP seeds was manually dissected in extraction buffer from the DNAeasy kit (Qiagen). We followed the protocol for DNA extraction as recommended in the manual. Isolated DNA was sent directly to Novogene (Hongkong, China) for whole genome or bisulfite sequencing on a HiSeqX in 150-bp paired-end mode. Two replicates of 4DAP endosperm bisulfite sequencing were performed for each cross: *Cr × Cg*, *Cr × Cr*, *Cg × Cg*.

### RNAseq analyses

For RNA analysis, reads were mapped to the *Arabidopsis* or the *Capsella* reference genomes, using TopHat v2.1.0 [74]. Gene expression was normalized to reads per kilobase per million mapped reads (RPKM) using GFOLD [75]. Expression level for each condition was calculated using the mean of the expression values in both replicates. Differentially regulated genes across the replicates were detected using the rank product method, as implemented in the Bioconductor RankProd Package [76].

### Variant frequency calling

Whole genome sequencing data from 4DAP manually dissected endosperm from *Cr × Cr* and *Cr × Cg* were mapped to the *Cr* reference genome. Variants were called with bcftools call/mpileup [77] (http://samtools.github.io/bcftools/bcftools.html). Only variants specifically present in the *Cr × Cg* data were considered to be *Cg* variants. Means of the ratio between *Cg* variant/base coverage were calculated for each chromosome.

### CENH3 antibody design

Custom antibodies for detection of CENH3 from *Capsella* were generated by BioNordika AB. Two peptides were used for immunization of rabbits: NH2- C+DFDLARRLGGKGRPW–COOH and NH2- C+QASQKKKPYRYRPGT–CONH2. Antibodies were subjected to affinity purification and validated by ELISA. They were coupled to KLH (keyhole limpet hemocyanin) carrier protein and delivered in standard buffer (PBS 1x, 0.01% thimerosal and 0.1% BSA).

### Quantitative data

Raw data for all plots in the manuscript are shown in S1 Data.

## Supporting information

**S1 Fig. Centromeric repeats in *Cr* and *Cg* genomes are identical.** Frequency of SNPs along centromeric repeats in *Cr* and *Cg* genomes.
(EPS)

**S2 Fig. Protein alignment of *Cr* and *Cg* CENH3 variants and H3 variants.** Protein alignment of *Cr* and *Cg* CENH3 variants, H3.1 (Carubv10003405m and Cagra.4395s0104 are identical to each other) and H3.3 (Carubv10021126m and Cagra.0799s0032.1 are identical to each other). Red arrows indicate amino acid differences between the proteins. Grey areas mark the sequences used as peptides for antibody production.
(EPS)

**S3 Fig. Methylation level of genes in *Cr*, *Cg* and *Cr* × *Cg* endosperm of 4 DAP seeds.** A) Metagene plots showing methylation level of genes in *Cr*, *Cg* and *Cr* × *Cg* endosperm of 4 DAP seeds. B) Boxplots showing the methylation level of genes in *Cr*, *Cr* × *Cg* and *Cg* endosperm of 4 DAP seeds. Boxes show medians and the interquartile range, and error bars show the full range excluding outliers. Asterisks indicate significant differences calculated by Wilcoxon test (*** p-value < 0.001).
(EPS)

**S4 Fig. Upregulated genes in hybrid endosperm are preferentially localized in pericentromeric regions.** A) Number of genes upregulated in *Cr* × *Cg* in comparison to *Cr* and *Cg* per decile of genes on each scaffold. Pericentromeric regions (p) are highlighted in grey for each scaffold B) Number of genes downregulated in *Cr* × *Cg* in comparison to *Cr* per decile of genes on each scaffold. Pericentromeric regions (p) are highlighted in grey for each scaffold. C) Venn diagram showing the overlap between significantly overexpressed genes in hybrid seeds compared to *Cr* × *Cr* and *Cg* × *Cg* and PHE1 targets (p-values were calculated using the supertest function from R package SuperExactTest [78]).
(EPS)

**S5 Fig. *AGL40* and *AGL95* are overexpressed in *Capsella* hybrid endosperm and in the endosperm of Arabidopsis triploid seeds.** A) Expression level of *AGL95* and *AGL40* in *Cr*, *Cg* and *Cr* × *Cg* endosperm. *B)* Expression level of *AGL95* and *AGL40* in the endosperm of diploid and triploid seeds. C) CHG and CHH methylation on *AGL40*. D) CHG and CHH methylation on *AGL95*.
(EPS)

**S6 Fig. PHE1 target genes are overexpressed in *Capsella* hybrid endosperm and in the endosperm of *Arabidopsis* triploid seeds.** Scatter plot showing expression of PHE1 target genes in *Cr* × *Cg* compared to *Cr* × *Cr* seeds and deregulated genes in the endosperm of *Arabidopsis* triploid versus diploid seeds. Upregulated PHE1 target genes are highlighted in red (p value was calculated using the Pearson correlation test).
(EPS)

**S1 Table. Localization of TEs loosing DNA methylation.**
(EPS)

**S1 Data. Raw data for all quantitative assays shown in the manuscript.**
(XLSX)

## Acknowledgments

We thank Kim Steige for helping with generating the F2 population used in this study.

## Author Contributions

**Conceptualization:** Katarzyna Dziasek, Lauriane Simon, Clément Lafon-Placette, Tanja Slotte, Claudia Köhler.

**Data curation:** Benjamin Laenen, Juan Santos-González.

**Formal analysis:** Lauriane Simon, Benjamin Laenen, Juan Santos-González.

**Funding acquisition:** Claudia Köhler.

**Investigation:** Katarzyna Dziasek, Lauriane Simon, Clément Lafon-Placette, Cecilia Wärdig.

**Methodology:** Tanja Slotte, Claudia Köhler.

**Project administration:** Claudia Köhler.

**Supervision:** Claudia Köhler.

**Validation:** Katarzyna Dziasek, Lauriane Simon, Claudia Köhler.

**Writing – original draft:** Katarzyna Dziasek, Lauriane Simon, Claudia Köhler.

**Writing – review & editing:** Katarzyna Dziasek, Lauriane Simon, Clément Lafon-Placette, Tanja Slotte, Claudia Köhler.

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
