## [Decision Letter · Decision Letter 0]

7 Oct 2020

Dear Dr SIMON,

Thank you very much for submitting your Research Article entitled 'Hybrid seed incompatibility in Capsella is connected to chromatin condensation defects in the endosperm' to PLOS Genetics. Your manuscript was fully evaluated at the editorial level and by independent peer reviewers. The reviewers appreciated the attention to an important problem, but raised some substantial concerns about the current manuscript. Particularly, serious concerns were raised by Referee 2 over the novelty of the findings presented in the manuscript. Based on the reviews, we will not be able to accept this version of the manuscript, but we would be willing to review again a revised version. We cannot, of course, promise publication at that time.

Should you decide to revise the manuscript for further consideration here, your revisions should address the specific points made by each reviewer. To address the concerns of Referee 2 and strengthen the main conclusions of the paper, we suggest additional revisions: 

1. The hypothesis of limited maternal sRNAs being the cause of heterochromatin decondensation in hybrid seed endosperm is novel and interesting, as pointed out by Referee 2. Additional data supportive of this idea would significantly strengthen the paper. Ideally sRNA sequencing should be performed. However, we understand this might be difficult given the input requirement of sRNA sequencing and the difficulty of endosperm isolation, especially at this challenging time. Therefore, we propose to perform in-depth analysis of existing methylation data to strengthen the link between sRNAs, DNA hypomethylation and heterochromatin decondensation. The authors find TE CHG/CHH methylation reduced in the hybrid endosperm. Detailed analysis could be performed to examine the methylation reduction over heterochromatic vs euchromatic TEs, CMT3/2- vs RdDM- governed TEs, 24-nt sRNA enriched vs depleted TEs, TEs that are in the pericentromeric regions or on chromosome arms, etc. 

2. A novel link was made between gene misregulation and heterochromatin hypomethylation. Pheres1 target genes localized in the pericentromeric region were found preferentially upregulated in hybrid seed, which is interesting but not fully supported by the data. The misregulated genes were identified from Cr x Cr and Cr x Cg comparison, which may be influenced by genetic differences between Cr and Cg. Therefore, Cg x Cg endosperm transcriptome should also be included in the analysis. If available, data from reciprocal crosses should also be analyzed to narrow down the causal genes. This analysis might establish more clearly the preferential upregulation of pericentromeric Pheres1 targets (Line 233). Additionally, DNA methylation at AGL95 and 40 should be shown in the hybrid Cr x Cg endosperm and the Cr and Cg parental endosperm.

3. The authors find Cg contains more centromeric repeat copies than Cr, which, however, does not lead to CENH3 loading problems in hybrid seed. It is important to show that the mitotic problems are not related to the centromere and CENH3. However, the speculation (line 319-320) “that the cause for the [pericentromeric] hypomethylation is an imbalance of maternal Cr 24-nt sRNAs and Cg centromeric repeats” is confusing. How centromeric repeat number can influence pericentromeric heterochromatin is unclear. Furthermore, data on the specificity of the Capsella CENH3 antibody should be included.

If you decide to revise the manuscript for further consideration at PLOS Genetics, please aim to resubmit within the next 60 days, unless it will take extra time to address the concerns of the reviewers, in which case we would appreciate an expected resubmission date by email to plosgenetics@plos.org.

[LINK]

We are sorry that we cannot be more positive about your manuscript at this stage. Please do not hesitate to contact us if you have any concerns or questions.

Yours sincerely,

Xiaoqi Feng, DPhil

Guest Editor

PLOS Genetics

Wendy Bickmore

Section Editor: Epigenetics

PLOS Genetics

Reviewer's Responses to Questions

**Comments to the Authors:**

Reviewer #1: The manuscript by Lauriane Simon et al provides an interesting perspective on the way endosperm development, but not embryo development induces mitotic abnormalities. They link this pattern of development to defaults to both CHG and CHH methylation, and suggest that PHE1 which acts as a transcription factor, might play an important role in the hyper activation of several relevant targets within pericentromeric heterochromatin. While the manuscript is well written and quite interesting, it suffer from a number of limitations that would need to be addressed to validate the core conclusions.

* importantly, the authors need to clarify how many replicates were performed for the chromosome spread in Fig 2A, B and C. I have not seen any numbers reported in the Material and Methods.

* Similarly, I was a bit confused with the number of replicates generated for the BS-seq analyses. While it may be difficult to generate enough samples from very early endosperm, the authors should provide the number of replicates generated. That would strengthen their argument.

* I’m also a bit confused with the fact that they did not observed any differences in the methylation level in genes (Fig S3), but report statistically significant differences on S3-B. How does that fit with the expected pattern ? I might have missed something, but I find it confusing.

* Surprisingly, CENH3 does not seem to have much impact of the compaction of chromocenters. Have the authors looked more closely, with enough replicates, to have enough certainty that CENH3 indeed does not account for chromosome loss ?

* While the data on TEs is quite interesting, Wilcoxon’s test most likely over-estimates the actual differences, the spread being rather broad. Student-test might be more appropriate.

* Also quite important, I could not understand how the DNA methylation analyses were performed on the PHE1 targets orthologs. While the promoter region is clearly important for DNA methylation, I could not really understand how the significance was calculated, and how the authors compared the PC versus NonPC targets. Did they average the mean in the CG, CHG and CHH context ? This needs to be clarified in the discussion, possibly adding the methodology used to compute the statistical tests.

minor point:

line 125 «Hybrid endosperm shows mitotic problems and random chromosome loss » should read « Hybrid endosperm shows mitotic abnormalities and random chromosome loss »

Reviewer #2: Review of ms entitled “Hybrid seed incompatibility in Capsella is connected to chromatin condensation defects in the endosperm” by Dziasek et al.

The work presented here seeks to better understand the poorly understood underlying mechanisms provoking seed abortion during hybridization of two recently diverged brassicaceae species.

Authors perform reciprocal crosses between Capsella rubella (referred to as Cr)  and Capsella grandiflora (referred to as Cg) and focus on studying the fate of the endosperm in the resulting hybrid seeds. Indeed, it is known that although both species have the same ploidy level, they are separated by a strong endosperm-based hybridization barrier (i.e. seed development is in part strongly perturbed as a result of poor endosperm development).

A QTL analysis identified three QTLs associated with Cr x Cg incompatibility. These QTLs are localized in pericentromeric regions, i.e. regions of condensed chromatin. No specific genes where identified, rather extended pericentromeric DNA regions.

Furthermore, authors found that in hybrid seeds:

1) the hybrid endosperm has mitotic problems and displays random chromosome loss. Authors convincingly show that chromosome loss is not attributable to the centromere-specific variant of H3 (CENH3).

2) the chromatin in the hybrid endosperm is decondensed, which is associated with loss of DNA methylation.

3) deregulated genes in hybrid endosperm are mostly localized in pericentromeric regions, in particular authors found increased expression of PHERES1, encoding a TF previously shown to account for triploid seed arrest.

The finding that hybrid endosperm displays random chromosome loss and lower DNA methylation levels was already reported previously (references are provided by the authors). Furthermore, previous reports showed (using A. lyrata and A. thaliana) that hybridization can cause chromosome chromatin compaction defects (e.g. Zhu et al Genome Biology 2017). So in this regard the findings reported in this work are not substantially novel.

The novelty present in this report could be the proposition that lower DNA methylation (inducing decondensation and chromosome loss) is due to lower sRNAs provided by maternal tissues as a result of their lower pericentromic repeat copy number. But this was not tested. Authors propose that seed abortion is due to high PHERES1 expression but it might just as well be due the result of mitotic problems and/or chromosome loss.

As mentioned above, similar observations were done using crosses involving parents of different ploidy levels. In these cases, authors proposed different mechanisms accounting for seed abortion that include ideas for mechanisms similar to those proposed here (e.g. Batista et al 2019, Erdmann et al 2017) but also other ones (e.g. Dilkes et al. 2008). However, they were tested by genetic means.

Reviewer #3: In this manuscript from the Dziasek et al, the Köhler lab continue their work on the effects of hybridization on endosperm development and seed viability in Capsella species. Using genetic, epigenetic, genomic, and cytological tools, the authors show that crosses of C. rubella with C. grandiflora show problems with mitosis and chromosome conservation in the endosperm, that the endosperm in these hybrids has reduced cytosine methylation and chromatin condensation, and that many PHERES1 target genes in pericentromeric regions are upregulated (likely the explanation for endosperm arrest). The authors also identify 3 QTLs, located in pericentromeric regions, which are responsible for endosperm arrest in Cr x Cg crosses. And the authors present correlative evidence that the effect Cr x Cg crosses on the endosperm, similar to the effect of 2x x 4x crosses in Arabidopsis, is due to an increase in pericentromeric repeat regions in Cg compared to Cr.

I have only minor comments:

lines 29-30: the can't say that the embryo was not affected, because they weren't able to judge the phenotype of chromatin in the embryo, because it was less dense in the parents of the hybrid. They should remove the reference to the embryo in this sentence.

lines 32-34: The mention of 'chromosome loss' comes out of nowhere, and needs to be better explained. In the previous sentence they talked about loss of chromosome condensation, a different effect of hybridization, making this sentence doubly confusing.

A general comment about the abstract: it's not as well written as the author summary, which is very nice.

line 67: 'syncitium': it's more correct to refer to the endosperm as a coenocyte. for definitions see: https://en.wikipedia.org/wiki/Coenocyte

line 88: I think this should be '2x x 4x', not '4x x 2x'

lines 143-144: the authors should modify this sentence. They didn't show that chromatin decondensation and chromosome loss were restricted to the endosperm. They showed that these things occur in the endosperm, and not in the root. There were not able to quantify the embryo. And they didn't look at any other tissues (not necessary look at other tissues), but they should modify the sentence to say the chromosome effects don't occur in all tissues, or something like that.

lines 181-183: this sentence has the first phrase in the present tense (are), and the next phrase in past tense (contained).

line 209: losing is misspelled as 'loosing' (also in S1 Table).

line 216: 'were' instead of 'was'

line 299: 'are' instead of 'is'

line 319-320: The sentence 'We speculate that the cause for the hypomethylation is an imbalance of

maternal Cr 24-nt sRNAs and Cg centromeric repeats.' is very interesting, and provides a possible mechanistic explanation for the effect on the endosperm that they see. I think it would be good to include a sentence like this is the abstract, because it ties all of their results together.

**Have all data underlying the figures and results presented in the manuscript been provided?**

Reviewer #1: Yes

Reviewer #2: Yes

Reviewer #3: Yes

PLOS authors have the option to publish the peer review history of their article (what does this mean?). If published, this will include your full peer review and any attached files.

Reviewer #1: No

Reviewer #2: No

Reviewer #3: **Yes: **Stewart Gillmor

---

## [Decision Letter · Decision Letter 1]

15 Jan 2021

Dear Dr SIMON,

We are pleased to inform you that your manuscript entitled "Hybrid seed incompatibility in Capsella is connected to chromatin condensation defects in the endosperm" has been editorially accepted for publication in PLOS Genetics. Congratulations!

Yours sincerely,

Xiaoqi Feng, DPhil

Guest Editor

PLOS Genetics

Wendy Bickmore

Section Editor: Epigenetics

PLOS Genetics

Comments from the reviewers (if applicable):

Reviewer's Responses to Questions

**Comments to the Authors:**

Reviewer #1: The manuscript by Dzalsek et al is a significant improvement from the last version. The authors have included most of my comments in either the methods (number of replicates for the spreads in Fig 2ABC, 3 biological replicates; number of replicates for the CENH3, 3 biological replicates; number of replicates for the BS-seq experiments 2 for CrxCg, CrxCr, CgxCg) or the figure legends (Fig5). Thus, I would suggest that the paper be accepted in its current version.

Reviewer #3: I am satisfied with the authors' responses to my comments on the manuscript.

**Have all data underlying the figures and results presented in the manuscript been provided?**

Reviewer #1: Yes

Reviewer #3: Yes

PLOS authors have the option to publish the peer review history of their article (what does this mean?). If published, this will include your full peer review and any attached files.

Reviewer #1: No

Reviewer #3: **Yes: **Stewart Gillmor

**Data Deposition**

http://datadryad.org/submit?journalID=pgenetics&manu=PGENETICS-D-20-01346R1

**Press Queries**

---

## [Editor Report · Acceptance letter]

2 Feb 2021

PGENETICS-D-20-01346R1 

Hybrid seed incompatibility in Capsella is connected to chromatin condensation defects in the endosperm 

Dear Dr Köhler, 

We are pleased to inform you that your manuscript entitled "Hybrid seed incompatibility in Capsella is connected to chromatin condensation defects in the endosperm " has been formally accepted for publication in PLOS Genetics! Your manuscript is now with our production department and you will be notified of the publication date in due course.

With kind regards,

Alice Ellingham

PLOS Genetics

On behalf of:
